# GSDistill: A Unified Paradigm for Geometry- and Semantics-Aware Document Pretraining

## Abstract

We introduce a unified pretraining paradigm for document understanding, grounded in a probability-theoretic formulation of multi-positive alignment and hierarchical self-distillation, which operate as complementary principles under a single objective. Unlike prior modular approaches, our framework redefines document pretraining as multi-positive, layout- and semantics-aware stochastic alignment rather than a collection of heuristic recipes. The model employs two complementary alignment heads: a semantic head, aligning page-level embeddings with OCR-derived text spans, and a geometric head, aligning representations with compact "box-text" descriptors that capture class type and structural layout. Both heads are trained with a multi-positive InfoNCE objective that supports one-to-many correspondences, alleviating the text-body bias of single-positive CLIP-style training and delivering markedly improved zero-shot document retrieval accuracy. To further strengthen representation quality, we incorporate a teacher-student self-distillation module with local-global hybrid regularization, enforcing patch-level consistency, global invariance, and embedding diversity. The resulting backbone produces layout-aware, language-grounded document representations that not only accelerate convergence and achieve competitive state-of-the-art results on layout detection benchmarks but also produce structured, consistent page-level embeddings that are naturally compatible with large language models, opening a path to advanced document reasoning and question-answering (QA).

## 1 Introduction

Document understanding is at the intersection of natural language processing and computer vision and is the foundation of various widely applied applications, including information extraction, semantic retrieval, and layout analysis. Traditional approaches have a tendency to divide this task into specialized modules.

**Document retrieval models**, i.e., retrieval-augmented generation (Lewis et al., 2020) and dense passage retrieval (Karpukhin et al., 2020), are targeted at modeling semantic similarity between queries and text passages for enabling efficient search in large document collections.

**Page layout detection systems**, e.g., DocLayout-YOLO (Zhao et al., 2024) and LayoutLM series (Xu et al., 2020b;a; Huang et al., 2022), operate on document pages as structured images. They identify text blocks, tables, figures, and headers, and encode them with geometric context beneficial for downstream tasks.

**Vision encoder pretraining for documents** has produced specialized models that integrate text, layout, and visual cues at the page level. Architectures such as LayoutLMv3 (Huang et al., 2022), DocFormer (Appalaraju et al., 2021), and LiLT (Wang et al., 2022) exemplify this line of work, yielding strong representations for form understanding, document classification, and QA.

Despite these advances, most methods remain tied to specific goals and have limited transferability between tasks. Performance on a given domain does not carry over to others, largely due to either narrow pretraining goals or highly specialized architectures. More general adaptability in retrieval and layout-focused tasks is therefore an open and urgent challenge.

In this work, we propose a conceptual shift in document pretraining: we move beyond fragmented recipes toward a principled unification of **semantics**, **geometry**, and **self-distillation**. Our framework operationalizes this shift by coupling dual-head multi-positive contrastive learning with hierarchical vision self-distillation under a single probabilistic objective. The resulting pretrained model learns representations that are both layout-aware and language-grounded with smooth transferability to a wide range of downstream tasks without task-specific supervision. Moreover, this single model can work as a "one-stop" backbone for document intelligence through effortless adaptations to retrieval and layout analysis.

**Our contributions can be summarized as the following:**

- **Unified pretraining objective:** We formalize document pretraining as **multi-positive stochastic alignment**, where semantic and geometric signals are jointly optimized under dual heads. This reconceptualization establishes a new family of pretraining objectives rather than an incremental extension of CLIP-style training.

- **Integration of vision self-distillation:** We extend recent advances in self-distillation, specifically iBOT Zhou et al. (2021) and DINOv2 and DINOv3 (Oquab et al., 2023; Siméoni et al., 2025), which enforce patch- and page-level invariances in the vision backbone. These invariances strengthen the ability of the backbone to capture structural regularities while simultaneously improving the semantic grounding of document representations.

- **Layout detection:** We conduct evaluations on DocLayNet (Pfitzmann et al., 2022), PubLayNet (Zhong et al., 2019), and additional benchmark datasets. Our results show that fine-tuning only the detection head that is added to the pretrained backbone achieves performance that can level with or exceed current state-of-the-art systems.

- **Generalization under Subsampled Positives:** Our multi-positive InfoNCE loss enables strong zero-shot transfer in document retrieval, avoiding the characteristic long-context forgetting problem in single-pair contrastive models.

- **Flexible, scalable, and reproducible framework:** We implement the pretraining recipe in HuggingFace (Wolf et al., 2020) and Ultralytics (Jocher et al., 2023) frameworks with DeepSpeed Rasley et al. (2020) ZeRO-3 CPU-offloading optimization enabled for efficient training at scales. Our framework supports flexible combinations of self-distillation, masked patch prediction, and both single- and multi-positive training, with or without embedding-level regularization.

Overall, our work establishes a **general-purpose**, **layout-** and **language-aware** backbone that unifies retrieval and structural analysis within a single pretraining paradigm. More than an incremental recipe, it advances a design principle for document pretraining: modality-specific signals (semantics and geometry) must be decoupled yet jointly optimized under multi-positive alignment. This principle, coupled with hierarchical self-distillation, provides a scalable foundation for the next generation of document intelligence systems.

## 2 RELATED WORKS

### 2.1 MULTI-POSITIVE CONTRASTIVE LEARNING

Multi-positive contrastive learning (MPCL) extends the standard InfoNCE loss by allowing each anchor to align with multiple valid positives, mitigating the limitations of single-positive (SP) formulations. By averaging gradients across structured sets of positives, MPCL yields more stable supervision, alleviates semantic underfitting, and improves optimization dynamics. Its benefits have been demonstrated across domains: in NLP, SupMPN (Dehghan & Amasyali, 2022) leverages multiple hard positives and negatives for stronger sentence embeddings; in multilingual learning, MPCL (Zhao et al., 2023) exploits parallel translations for robust cross-lingual retrieval; in vision, multi-positive extensions (Liang et al., 2024) improve convergence and benchmarks such as CIFAR-10 (Krizhevsky & Hinton, 2009) and Tiny ImageNet (Le & Yang, 2015); in sensor-based activity recognition, MPSQCL (Ren et al., 2024) combines augmented views with quantum-boosted encoders; and in pose understanding and hierarchical retrieval, GenPoCCL (Inayoshi et al., 2024) and Hierarchical MPCL (Kavimandan et al., 2025) demonstrate domain-specific gains. While promising in noisy

and multimodal settings, MPCL remains unexplored for document pretraining, where one-to-many alignment is particularly relevant for capturing both semantic and structural cues.

## 2.2 SELF-DISTILLATION IN VISION MODELS

Self-distillation achieves strong representations without external supervision by enforcing consistency between a teacher and student network. Methods such as DINO (Caron et al., 2021; Oquab et al., 2023; Siméoni et al., 2025) and iBOT (Zhou et al., 2021) couple global invariance with patch-level prediction, producing robust and transferable features. Although widely validated on natural image benchmarks, systematic application to document understanding is limited, despite the domain's need for fine-grained layout sensitivity and holistic semantic coherence.

## 2.3 DOCUMENT RETRIEVAL

Dense retrieval models such as DPR (Karpukhin et al., 2020), ColBERT Khattab & Zaharia (2020), and large-scale retrievers (Lewis et al., 2020) have advanced text search, while multimodal contrastive learning, notably CLIP (Radford et al., 2021), demonstrated powerful image-text alignment. However, SP-based objectives assume each document anchor corresponds to a single canonical span, typically dominated by body text. This overlooks alternative signals—captions, headers, tables, and figures—leading to biased supervision and under-representation of secondary but important features. Document retrieval therefore demands multi-span, multi-positive alignment strategies to fully capture page-level semantics.

## 2.4 LAYOUT DETECTION

Layout analysis remains a foundational task in document intelligence. Datasets such as PubLayNet (Zhong et al., 2019), DocLayNet (Pfitzmann et al., 2022), and DocBank (Li et al., 2020) have spurred progress from R-CNN based detectors to efficient YOLO variants (Zhao et al., 2024; Xu et al., 2020b;a; Kim et al., 2022). While these systems achieve strong detection and segmentation, they are largely specialized to local element classification. Their representations are not readily transferable to higher-level document understanding tasks such as retrieval or QA.

## 3 HIGH-LEVEL ARCHITECTURE

Figure 1 shows the proposed pretraining framework. The architecture is modular in nature and integrates multi-positive contrastive learning, self-distillation, and regularization within a unified training paradigm for learning document representations.

The base of the framework is a Vision Transformer (ViT) encoder that processes raw document inputs and is the shared basis for downstream tasks. Two heads complement the ViT: a *semantic head* that projects page embeddings into alignment with text spans such as OCR tokens, captions, and section headings, and a *geometric head* that projects into alignment with layout descriptors from bounding boxes. The two heads are trained on content-based and layout-based multi-positive contrastive losses, respectively, to ensure the learned representation is semantically coherent and layout-aware.

To further stabilize training and reduce representation collapse, we generalize KoLeo regularization (Sablayrolles et al., 2018) to KoLeo-hybrid regularization, balancing global embedding dispersion and local semantic preservation.

Meanwhile, the teacher-student self-distillation module provides additional supervision. The teacher network is updated as an exponential moving average of the student, offering stable targets for multiple augmented views of a page. The student is trained to align these targets through a *multi-scale consistency loss*, which enforces invariance over global and local views, and a *patch-level masked prediction loss*, which encourages recovery of fine-grained structural cues. Together, these tasks encourage the encoder to learn both hierarchical content semantics and spatial layout regularities.

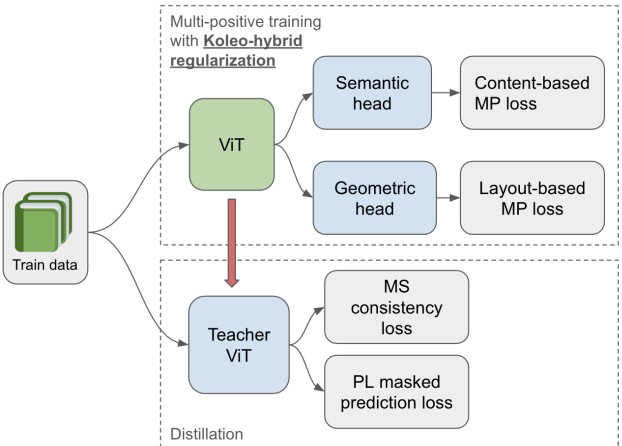

Figure 1: Overview of the proposed pretraining framework. The model integrates **multi-positive contrastive learning (MPCL)** (Section 4.1), implemented with dual semantic and geometric heads and enhanced through **KoLeo-hybrid regularization** (Section 4.5), to jointly encode content and layout information. In parallel, a **self-distillation module** with a teacher-student ViT enforces cross-view consistency via **multi-scale consistency loss** (Section 4.3) and **patch-level masked prediction loss** (Section 4.4). Together, these objectives encourage the model to acquire semantically coherent, layout-aware, and transferable representations that support a broad spectrum of document understanding tasks.

## 4    OBJECTIVE LOSS FUNCTION

Our pretraining framework is directed by a combined objective that blends **dual-head multi-positive contrastive learning (MPCL)**, **self-distillation**, and **regularization**. Our framework design explicitly encodes the semantic content and geometric structure of documents, along with enforcing cross-view consistency. The result is a set of powerful and transferable representations that go beyond localized detection and generalize to a broad variety of document understanding tasks.

### 4.1    MULTI-POSITIVE CONTRASTIVE LEARNING

Let $h_p \in \mathbb{R}^d$ denote the embedding of a document page, and let $\{z_1, z_2, \ldots, z_K\}$ represent a set of $K$ semantically consistent positives. Depending on the head, these positives correspond to:

- **Semantic head:** OCR text spans, titles, and captions ($z_i^{\text{sem}} = f_{\text{sem}}(t_i)$).

- **Geometric head:** box-text descriptors encoding class type and bounding box coordinates ($z_i^{\text{geom}} = f_{\text{geom}}(b_i)$).

In contrast to standard InfoNCE, which assumes a single positive, MPCL distributes supervision across all valid matches. The objective for anchor $h_p$ is:

We first define the *multi-positive contrastive loss (MPCL)* for a page representation $h_p$ and its set of $K$ positive spans $\{z_i\}_{i=1}^{K}$:

$$\mathcal{L}_{\text{MPCL}}(h_p, \{z_i\}) = -\log \frac{\sum_{i=1}^{K} \exp(\text{sim}(h_p, z_i)/\tau)}{\sum_{i=1}^{K} \exp(\text{sim}(h_p, z_i)/\tau) + \sum_{j=1}^{N} \exp(\text{sim}(h_p, n_j)/\tau)},$$

where $\text{sim}(\cdot, \cdot)$ denotes cosine similarity, $\tau$ is the temperature hyperparameter, and $\{n_j\}_{j=1}^{N}$ are negatives sampled from the batch or a memory bank. Unlike the standard SP loss, this formulation aggregates evidence across multiple valid alignments (e.g., captions, OCR spans, section headers), thereby:

- **Stabilizing gradients**: Averaging over multiple positives reduces variance in the learning signal and mitigates the risk of noisy or ambiguous correspondences dominating the update, see Figures A1 and A2 in the Appendix for the curves of gradient norms and loss curves at training.

- **Balancing supervision across elements**: By treating multiple valid alignments as equally weighted positives, the model avoids over-emphasizing dominant regions (e.g., long paragraphs or large visual blocks) at the expense of smaller but semantically important elements such as captions, equations, or footnotes. This balanced training signal ensures that representations capture both major structures and subtle details, leading to embeddings that are less biased by element size and more faithful to the full semantic variety of a document page.

## 4.2 DUAL-HEAD EXTENSION

We treat the dual-head MPCL not as an ad-hoc extension, but as a **structural necessity**. When semantic and geometric cues are fused directly, they often compete and degrade retrieval; when decoupled under multi-positive alignment, they instead provide complementary supervision. This motivates the dual-head objective:

$$\mathcal{L}_{\text{dual}} = \mathcal{L}_{\text{sem}}(h_p, \{z_i^{\text{sem}}\}) + \mathcal{L}_{\text{geom}}(h_p, \{z_i^{\text{geom}}\}).$$

Here, $\mathcal{L}_{\text{sem}}$ aligns the page embedding with semantic positives (*e.g.*, OCR text spans, captions, headers), while $\mathcal{L}_{\text{geom}}$ aligns it with geometric positives (*e.g.*, layout boxes, structural anchors). This separation ensures that the learned representation remains both **semantically coherent** and **geometrically grounded**, a property essential for document understanding and broadly applicable to any setting with heterogeneous alignments.

## 4.3 MULTI-SCALE CONSISTENCY SELF-DISTILLATION

Beyond the dual-head contrastive objectives, we incorporate **self-distillation** to enhance representation quality. Following DINOv3 (Siméoni et al., 2025), we maintain a momentum-encoder teacher that produces stable targets for both global and local views, while the student is trained for consistency across multiple augmentations. Formally, given global views $\{v_g\}$ and local crops $\{v_\ell\}$, the teacher outputs $y_g, y_\ell$ and the student predicts $p_g, p_\ell$. The distillation loss is:

$$\mathcal{L}_{\text{distill}} = \sum_{v \in \{v_g, v_\ell\}} \text{CE}(p_v, y_v),$$

where CE is cross-entropy. Teacher parameters are updated as an exponential moving average (EMA) of the student, providing stable supervision.

In contrast to classical DINO, where the teacher supervises only global crops, we extend the teacher to also predict local views, following the iBOT paradigm. This design enforces multi-scale consistency: at the global level, it aligns page-level structures such as layout topology and topic arrangement, while at the local level, it strengthens representations of finer-grained regions including text lines, tables, figures, and captions. By jointly aligning global and local signals, the model acquires hierarchy-aware features that are robust to distortions and partial observations, a property particularly beneficial for downstream layout analysis and multimodal alignment in complex documents. Ablation results from Table A1 in the Appendix demonstrates the effectiveness of our choice.

## 4.4 PATCH-LEVEL MASKED PREDICTION

To complement global and local consistency, we adopt the **iBOT masked prediction objective**. Let $\mathcal{M}$ be the set of masked patches, and let $p_m$ and $y_m$ denote the student prediction and teacher target for $m \in \mathcal{M}$. The loss is:

$$\mathcal{L}_{\text{ibot}} = \frac{1}{|\mathcal{M}|} \sum_{m \in \mathcal{M}} \text{CE}(p_m, y_m),$$

This objective enforces patch-level consistency, encouraging recovery of fine-grained structural cues beyond the holistic page embedding and complementing global–local self-distillation.

For document understanding, patch-level prediction strengthens representations of localized structures such as text lines, tables, figures, and captions. By training the model to restore masked content, it becomes more robust to noise, partial page views, and irregular layouts, which are frequent in scanned and multilingual documents.

### 4.5 KoLeo-Hybrid Regularization

A key challenge in dual-head MPCL is avoiding representation collapse under strong alignment pressures. To address this, we introduce a **KoLeo-Hybrid regularizer** that balances global uniformity with local semantic coherence. Unlike prior uses of KoLeo (Sablayrolles et al., 2018) in vision-only self-supervised learning, our adaptation is tailored for document pretraining: semantic neighborhoods are constructed through multi-view page augmentations, ensuring intra-page consistency while maintaining inter-page discrimination.

Formally, the loss is:

$$\mathcal{L}_{\text{KoLeo-H}} = \alpha \cdot \log \mathbb{E}_{i \neq j} \left[ \exp\left(\text{sim}(\bar{h}_i, \bar{h}_j)\right) \right] + \beta \cdot \frac{1}{|\mathcal{P}|} \sum_{(u,v) \in \mathcal{P}} \left(1 - \text{sim}(\bar{h}_u, \bar{h}_v)\right), \tag{1}$$

where $\bar{h}$ denotes $L_2$-normalized embeddings and $\mathcal{P}$ denotes the set of semantically related pairs. The coefficients $\alpha$ and $\beta$ control the trade-off between dispersion and preservation. We apply this regularizer to **both semantic and geometric embeddings**, so that each head maintains globally diverse geometry while preserving local coherence within its modality. This hybridization allows the model to resist collapse, retain task-relevant semantic and structural structure, and scale robustly under multi-positive alignment.

### 4.6 Full Objective

The final pretraining objective is a weighted sum of all components:

$$\mathcal{L} = \lambda_1 \mathcal{L}_{\text{sem}} + \lambda_2 \mathcal{L}_{\text{geom}} + \lambda_3 \mathcal{L}_{\text{distill}} + \lambda_4 \mathcal{L}_{\text{ibot}} + \lambda_5 \mathcal{L}_{\text{KoLeo-H}},$$

where $\lambda_i$ are tunable coefficients controlling the trade-off.

This composite formulation ensures that the model learns **multi-positive, layout-aware, semantically faithful, and structurally consistent** document representations, enabling a single backbone to support detection and retrieval, see Table A2 in the Appendix for details on training hyperparameters.

## 5 Pretraining Data Format

Our pretraining framework requires data representations that express both the **semantic content** and the **structural geometry** of documents. To this end, each page of a document is divided into three complementary modalities: raw image views, text spans retrieved via OCR and captions, and box-text descriptors that encode geometric structure.

### 5.1 Page Images and Views

The page image serves as the primary visual input to the backbone. For self-distillation, we follow the DINOv3 recipe and generate multiple augmented views of each page:

- **Global views:** resized crops covering the entire page, preserving holistic context.
- **Local views:** random smaller crops that emphasize specific regions such as tables, figures, or headers.

These views are incorporated into the dataset alongside the original page, ensuring that both global and local perspectives are explicitly available during training. They are then used for the teacher-student consistency loss (Section 4.3), the patch-level consistency loss (Section 4.4), and the hybrid

regularization (Section 4.5). This design enforces invariance at the global level while preserving sensitivity to fine-grained layout cues, resulting in representations that are robust, semantically rich, and layout-aware.

## 5.2 SEMANTIC SUPERVISION FROM OCR AND CAPTIONS

To obtain element-level text supervision, we perform OCR with Gemini-2.0-flash (DeepMind, 2024) on each page and extract all text spans (paragraphs, headers, footnotes, etc.), as well as any surrounding captions for figures and tables. We encode each text element $t_i$ with a text encoder $f_{\text{sem}}(\cdot)$ into an embedding $z_i$. The resulting set of positives $\{z_1^{\text{sem}}, z_2^{\text{sem}}, \dots, z_K^{\text{sem}}\}$ forms the target distribution for the **semantic head** in our multi-positive contrastive loss.

## 5.3 GEOMETRIC SUPERVISION FROM BOX-TEXT DESCRIPTORS

In addition to textual contents, we construct lightweight **box-text descriptors** of geometric layout attributes. For each region annotation $b_i$ (e.g., paragraph, table, figure, caption), we have a short text string summarizing its *type* (class label) and geometric attributes (coordinates). These descriptors are fed to the same text encoder to generate geometric embeddings $z_1^{\text{geom}}, \dots, z_K^{\text{geom}}$. The **geometric head** maps the page embedding to this set under the same multi-positive contrastive framework, learning structural relationships between regions.

## 5.4 PRETRAINING TUPLE

Overall, as portrayed in Figure 2, each page is represented as a structured tuple:

$$\mathcal{D} = \{I_p, \ \{v_{g_j}\}_{j=1}^{G}, \ \{v_{\ell_j}\}_{j=1}^{L}, \ \{t_i\}_{i=1}^{K}, \ \{b_i\}_{i=1}^{K}\},$$

where $I_p$ is the original page image, $\{v_j\}$ denotes the set of global and local augmented views derived from $I_p$, $\{t_i\}$ are textual spans, and $\{b_i\}$ are box-text descriptors. During training, $\{v_g, v_\ell\}$ provide inputs for regularized self-distillation, while $\{t_i\}$ and $\{b_i\}$ serve as multi-positive sets for the semantic and geometric contrastive heads. This unified data format enables a single backbone to jointly learn visual, semantic, and structural representations of documents, ensuring consistency across both global context and fine-grained layout cues. Due to limited resources, we managed to collect and annotate 600K such tuples.

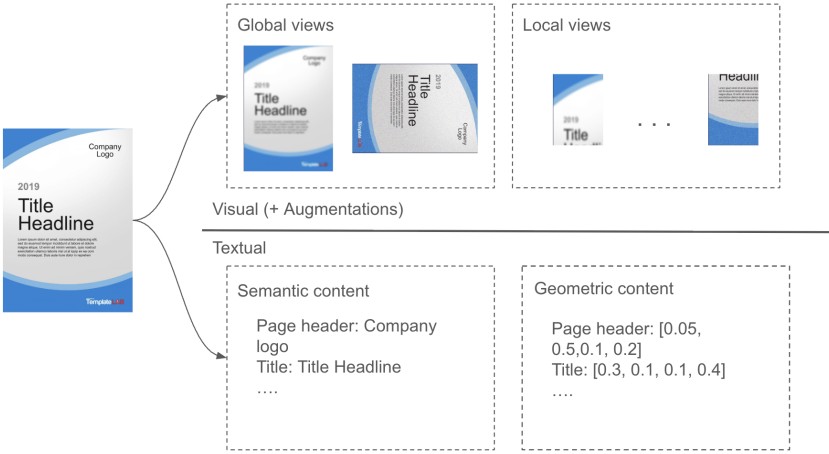

Figure 2: Illustration of the pretraining tuple $\mathcal{D}$. Each document page is represented by (i) the raw page image $I_p$, (ii) global and local augmented visual views $\{v_g, v_\ell\}$ used for self-distillation, (iii) semantic spans $\{t_i\}$ such as headers, titles, and captions, and (iv) geometric descriptors $\{b_i\}$ encoding bounding-box coordinates and region types. This unified representation provides a consistent input format for contrastive heads and distillation objectives, enabling the backbone to jointly learn visual, semantic, and structural cues.

# 6 EXPERIMENTS

We evaluate our pretrained backbone on three complementary axes: (1) fine-tuning for document layout detection, (2) subsampled document retrieval accuracy, and (3) ablations on the effect of multi-positive contrastive learning. All experiments are conducted on widely used document understanding benchmarks.

## 6.1 LAYOUT DETECTION

We assess our pretrained backbone (ViT-24, 70.2M) on PubLayNet and DocLayNet using head-only finetuning (5.7M parameters). The backbone is frozen, and only detection heads are optimized. Despite the reduced active parameter size and the lack of task-specific supervision, Table 1 shows

Table 1: Comparison of layout detection performance (mAP@50 and mAP@[50:95]) on PubLayNet and DocLayNet. Baseline SOTA numbers are taken from prior literature; DocLayout-YOLO results from Zhao et al. (2024); YOLO-DocLayNet results from YOLO-DocLayNet (2025); Hybrid approach results from Shehzadi et al. (2024).

| Dataset | Model | Trainable Params (M) | mAP@50 | mAP@[50:95] |
|---|---|---|---|---|
| PubLayNet | **ViT-24 (YOLOv10, head-only finetune)** | 5.7 | 97.9 | 94.0 |
| | Hybrid Approach | ≈55.0 | **98.8** | **97.3** |
| DocLayNet | YOLO-DocLayNet (YOLOv12n) | 2.6 | - | 75.6 |
| | **ViT-24 (YOLOv10, head-only finetune)** | 5.7 | **93.7** | 81.1 |
| | YOLO-DocLayNet (YOLOv12s) | 9.3 | - | 78.2 |
| | DocLayout-YOLO (YOLOv10m++) | ≈18.0 | 93.4 | 79.7 |
| | Hybrid Approach | ≈55.0 | 93.5 | **81.6** |
| | YOLO-DocLayNet (YOLOv12x) | 59.1 | - | 79.4 |

that our approach matches state-of-the-art detectors that are far larger and exceeds detectors that have around the same number of trainable parameters. Notably, a single epoch already yields strong results (84.1 mAP@50 on PubLayNet, 79.0 mAP@50 on DocLayNet; see Figure A3), with further gains from additional epochs. These results highlight the transferability of the dual-head pretraining to layout-aware detection.

## 6.2 SUB-SAMPLED DOCUMENT RETRIEVAL

In real retrieval settings,

Table 2: Document retrieval accuracy (%) with subsampled crops. Multi-positive training provides a clear advantage when only a single crop is available. As more crops are provided at inference, the gap narrows, and both methods converge. For the SP, crops are concatenated according to the original reading order. The results are obtained from 1,000 randomly sampled document pages that are not included in the 600K pretraining dataset.

| Method | 1 Crop | 5 Crops | 8 Crops |
|---|---|---|---|
| SP (Full Objective) | 45.2 | 61.8 | 66.3 |
| MPCL (Full Objective) | **54.6** | **68.1** | **71.2** |

supervision is often limited to captions, headers, or a few OCR spans rather than full text.

Formal proofs, grounded in probability theory, of how MPCL can increase retrieval accuracy are given in Sections S3 and S4 of the Appendix. CLIP-style one-to-one alignment tends to overfit dominant body text, neglecting secondary signals. By contrast, multi-positive contrastive learning aligns page embeddings with all consistent cues, improving robustness under sparse supervision. As shown in Table 2, this advantage is most pronounced when only limited signals are available, while performance gap tends to shrink once the full page semantics are increasingly accessible.

## 6.3 ABLATION STUDIES

We conduct ablation experiments to disentangle the contributions of semantic and geometric MPCL, self-distillation, and regularization. Table 3 reports results across seven configurations, from SP baselines to the complete system. MPCL is the primary driver of retrieval gains: semantic MPCL

Table 3: Ablation study on the contributions of MPCL, geometric alignment, self-distillation, and regularization.

| Configuration | Detection mAP (DocLayNet) | Retrieval mAP (1 Crop) |
| --- | --- | --- |
| SP (semantic) | 62.7 | 44.7 |
| SP (semantic + geometric) | 78.3 | 44.1 |
| SP (semantic ∥ geometric) | 65.3 | 42.9 |
| MPCL (semantic only) | 62.1 | 52.9 |
| MPCL (semantic + geometric) | 78.8 | 52.7 |
| MPCL (dual-head) + distill | 79.4 | 53.1 |
| MPCL (dual-head) + distill + ibot | 80.3 | 53.7 |
| MPCL (dual-head) + distill + ibot + KoLeo-H | **81.1** | **54.6** |

outperforms its SP counterpart (52.9 vs. 44.7). Geometric alignment consistently boosts detection (78.8 vs. 62.1 with MPCL; 78.3 vs. 62.7 with SP) but slightly reduces retrieval, reflecting a trade-off between spatial localization and text alignment. The failure of the concatenated variant highlights a key paradigm: naive fusion collapses retrieval, while dual-head MPCL preserves complementary structure. Adding self-distillation further improves stability and performance (79.4 detection, 53.1 retrieval), and ibot enhances patch-level consistency (+0.9 detection, +0.6 retrieval). Finally, KoLeo-Hybrid regularization ensures diversity under strong alignment pressures, yielding the best results: 81.1 detection and 54.6 retrieval.

## 7 CONCLUSION

In conclusion, we introduce a pretraining paradigm for document understanding that combines dual-head multi-positive contrastive learning, self-distillation, and hybrid regularization. The approach produces layout-aware, language-grounded representations that deliver improved zero-shot document retrieval and achieve competitive state-of-the-art detection performance with faster convergence. Beyond detection, the backbone provides a versatile foundation for OCR pipelines and for integration with large language models to perform more complex document understanding tasks such as reasoning and QA, establishing multi-positive alignment as a general paradigm for multi-modal document pretraining. Beyond detection and retrieval, this principle opens the door to OCR pipelines and LLM-based reasoning, providing a transferable design foundation rather than a task-specific recipe.

## LIMITATIONS AND FUTURE WORK

While our framework establishes a new paradigm, it has limitations. First, evaluation is restricted to PubLayNet and DocLayNet, leaving downstream reasoning and QA tasks for future work. Second, our pretraining data (600K tuples) is modest compared to billion-scale vision-language models, and scaling remains an open challenge. Third, OCR-based supervision inherits biases from current systems, limiting robustness in low-resource or multilingual settings. Addressing these limitations offers a natural path forward and will strengthen the generality of the paradigm. In addition, while our analysis is probability-theoretic, our training framework remains deterministic. We leave exploration of truly probabilistic approaches to future work

LLM USAGE STATEMENT

We acknowledge the use of large language models (LLMs) solely for text polishing and language refinement. All ideas, experiments, and analyses presented in this work are entirely the authors' own.

REPRODUCIBILITY STATEMENT

To ensure transparency and reproducibility, we will release our complete codebase, including data preprocessing scripts, training configurations, and the end-to-end training pipeline. The repository will be made publicly available on GitHub and will include detailed documentation, environment setup instructions, and example runs. This will enable other researchers to replicate our experiments and extend our framework for future studies.

ETHICS STATEMENT

This work complies with the ICLR Code of Ethics.[1] Our experiments use a combination of publicly available benchmarks (PubLayNet, DocLayNet) and self-annotated data. No human subjects or personally identifiable information are involved. We acknowledge that dataset biases may persist and encourage future work to examine fairness and representational balance. Code and training pipelines will be released to ensure reproducibility.

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

# A  APPENDIX

## S1  MORE ABLATION RESULTS AND HYPERPARAMETERS

Table A1: Ablation study on the contributions of global and local supervision vs. global supervision alone in self-distillation

| Configuration | Detection mAP (DocLayNet) | Retrieval mAP (1 Crop) |
|---|---|---|
| MPCL (dual-head) + distill (global) + ibot + KoLeo-H | 80.8 | 54.0 |
| MPCL (dual-head) + distill (global+local) + ibot + KoLeo-H | **81.1** | **54.6** |

Table A2: Training hyperparameters used in our experiments.

| **Category** | **Parameter** | **Value** |
|---|---|---|
| Core | Text model name | `Qwen3-Embedding-0.6B` |
| | Projection dimension | 256 |
| | Logit scale init | $\log(1/0.07)$ |
| | Image pooling | Mean |
| | Freeze vision | False |
| | Freeze text | True |
| | Disable MP | True/False |
| | Single-CLIP mode | True/False |
| | Distillation enabled | True/False |
| Multi-Positive Loss | $\lambda_1$ | 1 |
| | $\lambda_2$ | 1 |
| | MP temperature | 0.07 |
| | MP normalize | True |
| Self-distill/ iBOT | Self-distill output dim | 65,536 |
| | Self-distill hidden dim | 2048 |
| | $\lambda_3$ | 0.7 |
| | $\lambda_4$ | 0.7 |
| | $\lambda_5$ | 0.02 |
| | $t_{\text{stu,img}}$ | 0.1 |
| | $t_{\text{tea,img}}$ | 0.04 |
| | $t_{\text{stu,patch}}$ | 0.1 |
| | $t_{\text{tea,patch}}$ | 0.04 |
| | Sinkhorn iterations | 3 |
| | Momentum (base) | 0.996 |
| | Momentum (end) | 0.9995 |

## S2 TRAINING LOGS

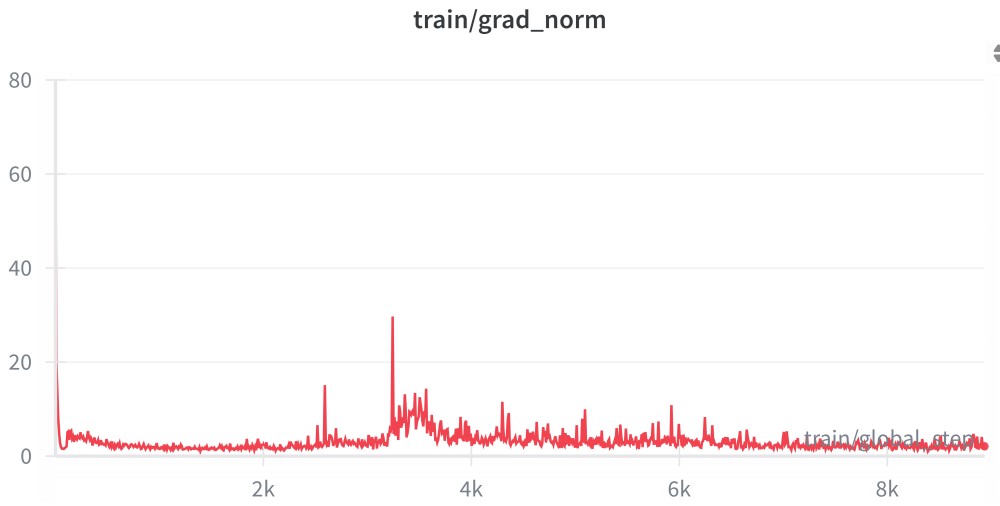

(a) Multi-positive contrastive learning (MPCL). The gradient norm of the full-stack loss remains smoother and more stable across training.

(b) SP objective. The gradient norm of the full-stack loss shows higher variance and less stability.

Figure A1: Gradient norms of the full-stack loss under different training objectives. **(a)** Multi-positive training yields smoother and more stable optimization dynamics, while **(b)** SP training exhibits noisier gradients. These results highlight the stabilizing effect of multi-positive supervision.

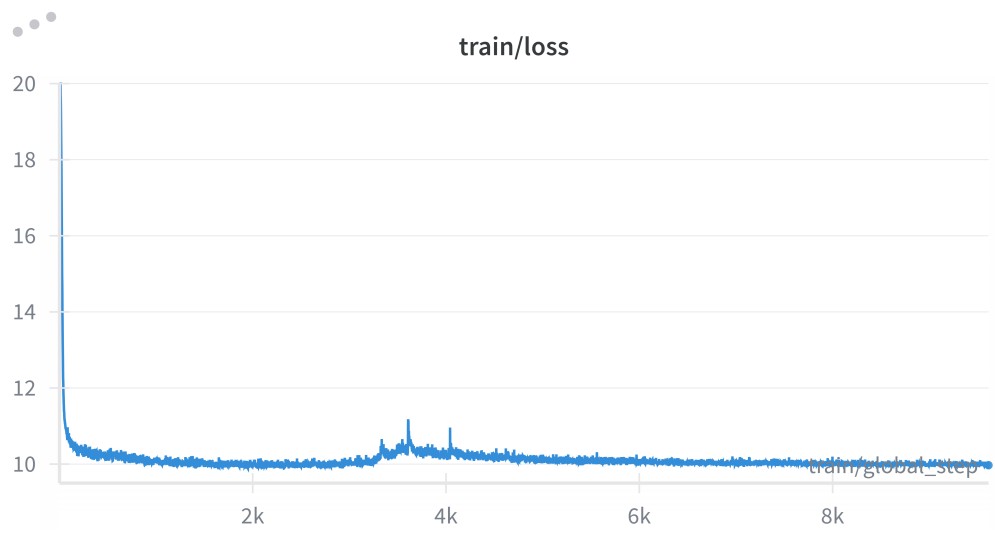

(a) Multi-positive contrastive learning (MPCL). The loss curve of the full-stack objective converges smoothly with reduced oscillations.

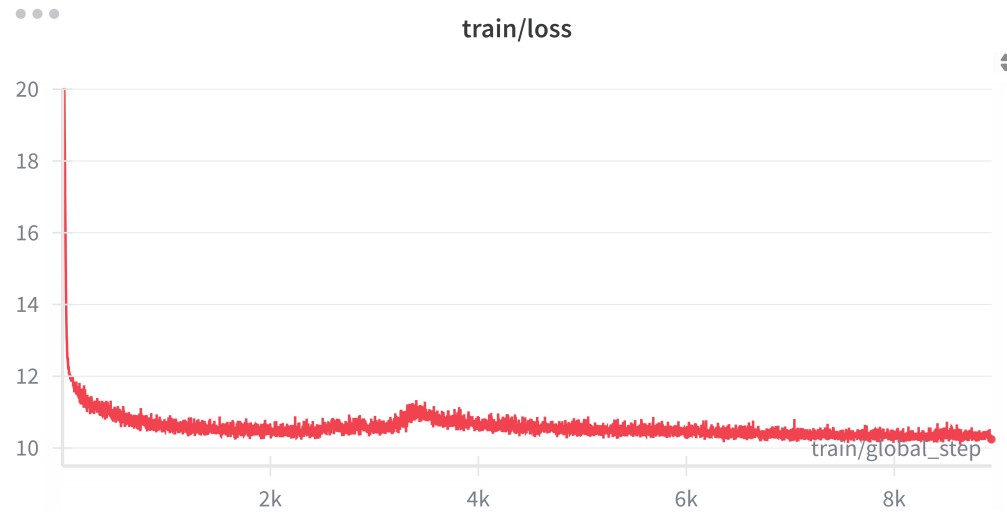

(b) SP objective. The loss curve of the full-stack objective exhibits higher variance and slower stabilization.

Figure A2: Training loss of the full-stack objective under different contrastive formulations. **(a)** Multi-positive training yields smoother convergence and reduced variance, while **(b)** the SP objective converges less stably. These results mirror the gradient norm analysis (Figure A1) and further highlight the stabilizing effect of multi-positive supervision.

The gap (MPCL loss being smaller) between the MPCL and SP loss curves are explained and proved in Lemma 5.

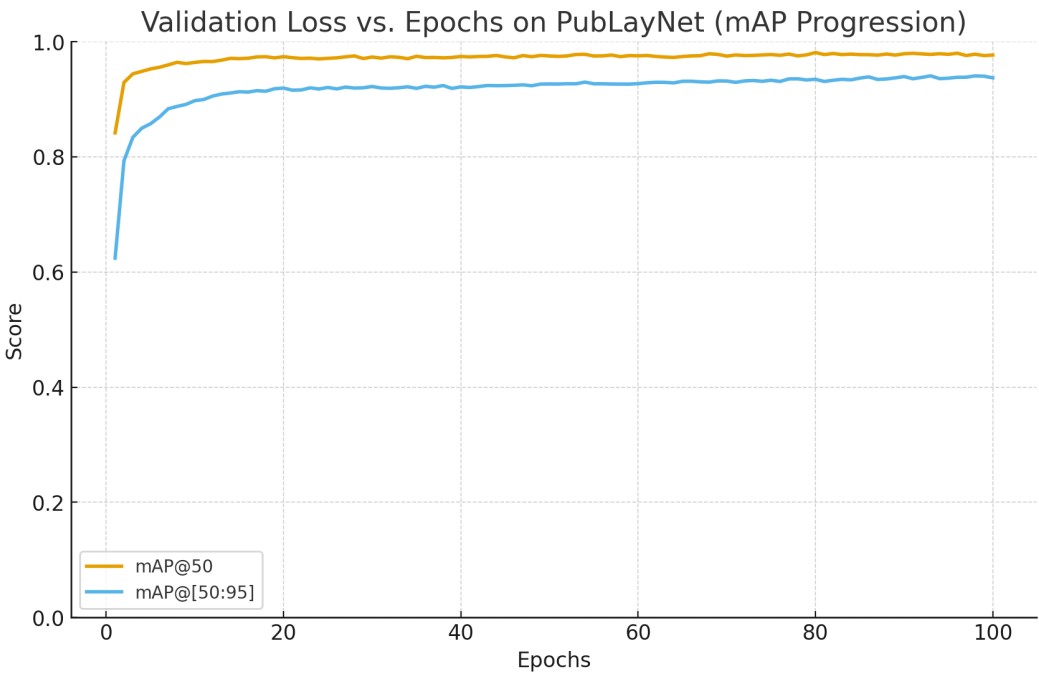

(a) Validation performance on PubLayNet with head-only finetuning. Remarkably, a single epoch is sufficient to reach high detection accuracy, with subsequent training yielding smoother convergence toward the final mAP scores.

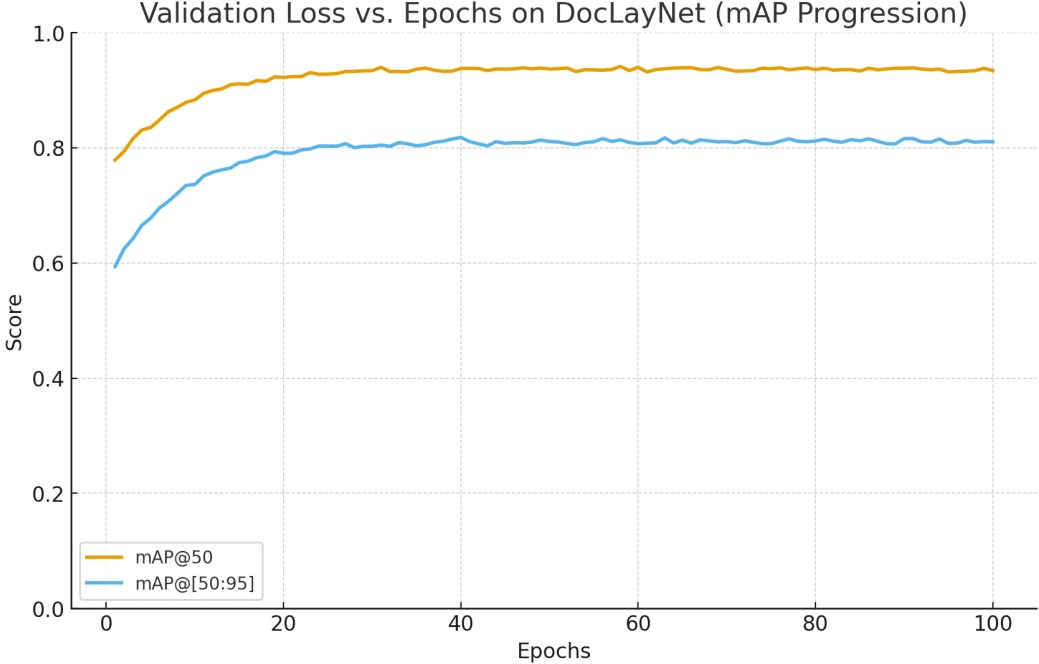

(b) Validation performance on DocLayNet with head-only finetuning. Even after one epoch, the model achieves strong baseline accuracy, though convergence is slower and exhibits larger variance compared to PubLayNet.

Figure A3: Head-only finetuning results on the 75M pre-trained backbone. In both **(a)** PubLayNet and **(b)** DocLayNet, a single epoch of training already produces competitive mAP scores, highlighting the strength of the pre-trained backbone. Additional epochs refine convergence: smoother and faster on PubLayNet, more fluctuating yet ultimately strong on DocLayNet.

## S3 A PROBABILISTIC SKETCH: MPCL IMPROVES RETRIEVAL PROBABILITY

**Goal.** We aim to show that *multi-positive contrastive learning (MPCL)* increases the probability of retrieving a correct document relative to *SP contrastive learning (SP)*.

**Setup.** Let a query (anchor) be $q$ and candidate documents score via $s(d) = \mathrm{sim}(q, d)$. The relevant set is $P = \{p_1, \ldots, p_m\}$ with i.i.d. scores $S_i = s(p_i) \sim F_p$, and the irrelevant set is $N = \{n_1, \ldots, n_K\}$ with i.i.d. scores $Z_j = s(n_j) \sim F_n$. Write $M_p^{(m)} = \max_{1 \leq i \leq m} S_i$ and $M_n = \max_{1 \leq j \leq K} Z_j$.

**Lemma 1** (Hit probability identity). *For any $m \geq 1$,*

$$\Pr(\text{hit} \mid m) = \Pr\{M_p^{(m)} > M_n\} = \mathbb{E}\big[1 - F_p(M_n)^{\,m}\big]. \tag{2}$$

*Proof.* By the law of total probability,

$$\Pr(M_p^{(m)} > M_n) = \mathbb{E}\Big[\Pr(M_p^{(m)} > M_n \mid M_n)\Big].$$

Conditioning on $M_n = t$ and using independence of positives and negatives,

$$\Pr(M_p^{(m)} > t \mid M_n = t) = 1 - \Pr(M_p^{(m)} \leq t).$$

Since $S_i$ are i.i.d. with CDF $F_p$, $\Pr(M_p^{(m)} \leq t) = [F_p(t)]^m$. Substituting and taking expectation over $M_n$ yields equation 2. $\qquad\square$

**Lemma 2** (Order-statistic advantage). *For any CDFs $F_p, F_n$ and $m \geq 1$,*

$$\mathbb{E}[1 - F_p(M_n)^{\,m}] \;\geq\; \mathbb{E}[1 - F_p(M_n)], \tag{3}$$

*with strict inequality unless $F_p(M_n) \in \{0, 1\}$ almost surely.*

*Proof.* For $a \in [0, 1]$ and $m \geq 1$, one has $a^m \leq a$. Applying this inside the expectation in equation 2 proves the claim. $\qquad\square$

**Theorem 1** (MPCL improves retrieval probability). *Suppose that the positive scores under MPCL stochastically dominate those under SP, $F_p^{\mathrm{MPCL}} \leq F_p^{\mathrm{SP}}$, and the negative scores are stochastically dominated, $F_n^{\mathrm{MPCL}} \geq F_n^{\mathrm{SP}}$. Then*

$$\Pr_{\mathrm{MPCL}}\Big\{M_p^{(m)} > M_n\Big\} \;\geq\; \Pr_{\mathrm{SP}}\Big\{M_p^{(1)} > M_n\Big\}.$$

*Proof.* By Lemma 1, hit probability is

$$\Pr(\text{hit} \mid m) = \mathbb{E}[g_{F_p}(M_n)], \quad g_{F_p}(t) = 1 - [F_p(t)]^m, \quad F_{M_n}(t) = [F_n(t)]^K.$$

(i) By monotonicity in $F_p$, if $F_p^{\mathrm{MPCL}} \leq F_p^{\mathrm{SP}}$, then by Lemma 2, $g_{F_p^{\mathrm{MPCL}}}(t) \geq g_{F_p^{\mathrm{SP}}}(t)$ pointwise, and thus

$$\int g_{F_p^{\mathrm{MPCL}}} \, dF_{M_n} \;\geq\; \int g_{F_p^{\mathrm{SP}}} \, dF_{M_n}.$$

(ii) By monotonicity in $F_n$, note that $g_{F_p}(t)$ is nonincreasing in $t$. If $F_n^{\mathrm{MPCL}} \geq F_n^{\mathrm{SP}}$, then

$$F_{M_n}^{\mathrm{MPCL}}(t) \geq F_{M_n}^{\mathrm{SP}}(t).$$

By the standard property of stochastic order, integration against a nonincreasing function preserves the inequality:

$$\int g_{F_p} \, dF_{M_n}^{\mathrm{MPCL}} \;\geq\; \int g_{F_p} \, dF_{M_n}^{\mathrm{SP}}.$$

(iii) Combining (i) and (ii), starting from $(F_p^{\mathrm{SP}}, F_n^{\mathrm{SP}})$ and moving along the path

$$(F_p^{\mathrm{SP}}, F_n^{\mathrm{SP}}) \;\to\; (F_p^{\mathrm{MPCL}}, F_n^{\mathrm{SP}}) \;\to\; (F_p^{\mathrm{MPCL}}, F_n^{\mathrm{MPCL}}),$$

yields

$$\Pr_{\mathrm{MPCL}}\Big\{\max_i S_i > \max_j Z_j\Big\} \;\geq\; \Pr_{\mathrm{SP}}\Big\{\max_i S_i > \max_j Z_j\Big\}.$$

$$\square$$

The two monotonicity assumptions are natural consequences of how MPCL modifies the learning dynamics relative to SP. First, if $F_p^{\mathrm{MPCL}}(t) \leq F_p^{\mathrm{SP}}(t)$ holds pointwise, then positive scores under MPCL are stochastically larger, meaning that true matches are less likely to receive low scores and more likely to achieve higher values. This reflects the intended effect of multi-positive training, which avoids penalizing alternative valid positives and therefore shifts the entire positive distribution to the right. Second, if $F_n^{\mathrm{MPCL}}(t) \geq F_n^{\mathrm{SP}}(t)$, then negative scores under MPCL are stochastically smaller, concentrating their mass at lower values and thus weakening the hardest competing negatives. This effect arises because MPCL alleviates false negatives, allowing the model to push genuine negatives further away. Together, these assumptions encode the intuitive separation that MPCL is designed to achieve, positives shift upward, negatives shift downward, and provide the stochastic dominance conditions under which improvements in retrieval probability follow directly.

## S4 MONOTONICITY PROPERTIES AND BASELINE ADVANTAGES OF MPCL

**Lemma 3** (Top-1 hit monotonicity in the number of positives)**.** *Let* $S_1, \ldots, S_m \overset{iid}{\sim} F_p$ *be positive scores and* $Z_1, \ldots, Z_K \overset{iid}{\sim} F_n$ *be negative scores, independent of* $\{S_i\}$*. Write* $M_n = \max_{1 \leq j \leq K} Z_j$ *and*

$$\mathrm{hit}_m \; = \; \Pr\big\{ \max_{1 \leq i \leq m} S_i > M_n \big\}.$$

*Then* $\mathrm{hit}_m$ *is nondecreasing in* $m$*. Moreover, if* $\Pr\{F_p(M_n) \in (0,1)\} > 0$ *(i.e., the setting is nondegenerate), the inequality is strict:* $\mathrm{hit}_{m+1} > \mathrm{hit}_m$ *for all* $m \geq 1$*.*

*Proof.* By Lemma 1, we have

$$\mathrm{hit}_m \; = \; \mathbb{E}[\, 1 - F_p(M_n)^m \,].$$

According to Lemma 2, $\mathrm{hit}_m$ is nondecreasing in $m$, and strictly increasing whenever $\Pr\{F_p(M_n) \in (0,1)\} > 0$. $\qquad\square$

A direct result from Lemma 3 is the following:

**Corollary 1** (MPCL baseline advantage under identical marginals)**.** *Suppose* $F_p^{\mathrm{MPCL}} = F_p^{\mathrm{SP}}$ *and* $F_n^{\mathrm{MPCL}} = F_n^{\mathrm{SP}}$*. Then for any* $m \geq 2$*,*

$$\Pr(\mathrm{hit}_m^{\mathrm{MPCL}}) \; \geq \; \Pr(\mathrm{hit}_1^{\mathrm{SP}}),$$

*with strict inequality in any nondegenerate setting where* $\Pr\{F_p(M_n) \in (0,1)\} > 0$*.*

**Lemma 4** (Softmax "win" probability is monotone in $m$)**.** *Let*

$$\pi_m \; = \; \mathbb{E}\left[ \frac{\sum_{i=1}^m e^{S_i/\tau}}{\sum_{i=1}^m e^{S_i/\tau} + \sum_{j=1}^K e^{Z_j/\tau}} \right], \qquad \tau > 0.$$

*Then* $\pi_m$ *is nondecreasing in* $m$*, with strict increase whenever* $\Pr\{\sum_{j=1}^K e^{Z_j/\tau} > 0\} > 0$ *and* $\Pr\{e^{S/\tau} > 0\} = 1$*.*

*Proof.* For any realization, write $A_m = \sum_{i=1}^m e^{S_i/\tau}$ and $B = \sum_{j=1}^K e^{Z_j/\tau} \geq 0$. Adding one more positive term $c = e^{S_{m+1}/\tau} > 0$ gives

$$\frac{A_m + c}{A_m + B + c} - \frac{A_m}{A_m + B} = \frac{cB}{(A_m + B)(A_m + B + c)} \; \geq \; 0,$$

with strict inequality whenever $B > 0$. Taking expectations preserves (strict) inequality, hence $\pi_{m+1} \geq \pi_m$ (strict if $B > 0$ with positive probability). $\qquad\square$

Lemmas 3 and 4, together with Corollary 1, formalize an important structural property of MPCL: retrieval performance improves monotonically with the number of positives, irrespective of whether the marginal score distributions differ from those of SP. This observation highlights that the benefits of multi-positive training are two-fold. Firstly, the stochastic dominance assumptions model distributional effects, positives pushed to the right and negatives to the left, ensuring improved separation in expectation. On the other hand, even without such distributional gains, aggregation over multiple

valid positives yields a combinatorial advantage: the maximum of $m$ independent draws stochastically dominates a single draw, and the softmax normalization similarly increases as more positive terms are included.

These observations imply that MPCL improves upon the "baseline" gain over SP irrespective of how much the score distributions shift during training. The monotonicity lemmas guarantee retrieval probability and softmax win probability are nondecreasing functions of $m$, establishing training objective robustness. Meanwhile, the corollary identifies that distributional shifts, if they exist, only reinforce this intrinsic advantage. Practically, this means the observed gains of MPCL can be understood as the superposition of structural gains from multi-positive aggregation and distributional gains from false negative mitigation.

## S5  MPCL vs. SP Loss Curves

**Lemma 5** (MPCL vs. SP under coupling). *Fix an encoder and a batch with positives $\{S_i\}_{i=1}^m$ and negatives $\{Z_j\}_{j=1}^K$. Let the SP positive $S$ be drawn uniformly from $\{S_i\}_{i=1}^m$ (coupling). Then pointwise,*

$$\log\Big(1 + \frac{\sum_j e^{Z_j/\tau}}{\sum_i e^{S_i/\tau}}\Big) \ \le\ \log\Big(1 + \frac{\sum_j e^{Z_j/\tau}}{e^{\max_i S_i/\tau}}\Big) \ \le\ \log\Big(1 + \frac{\sum_j e^{Z_j/\tau}}{e^{S/\tau}}\Big).$$

*Taking expectations yields $\mathbb{E}[\mathcal{L}_{\mathrm{MPCL}}] \le \mathbb{E}[\mathcal{L}_{\max}] \le \mathbb{E}[\mathcal{L}_{\mathrm{SP}}]$.*

*Proof.* Since $\sum_i e^{S_i/\tau} \ge e^{\max_i S_i/\tau}$, the middle term is larger than the left denominator, so the log decreases. Also $e^{\max_i S_i/\tau} \ge e^{S/\tau}$, so the right-hand log is largest. $\square$

**Remark 1.** *Lemma 5 assumes a coupling where MPCL and SP are evaluated on the same encoder outputs. This provides a structural inequality: $\mathbb{E}[\mathcal{L}_{\mathrm{MPCL}}] \le \mathbb{E}[\mathcal{L}_{\mathrm{SP}}]$. In practice, pretraining modifies the encoder differently under each objective, so their score distributions diverge. Nevertheless, the observed training curves consistently show lower MPCL losses, suggesting that the structural advantage persists even after accounting for distributional shifts.*

