# OpenReview forum: "GSDistill: A Unified Paradigm for Geometry- and Semantics-Aware Document Pretraining"
_ICLR.cc/2026/Conference — ICLR 2026 Conference Withdrawn Submission_

### Official Review · Reviewer_34mH · 2025-10-25

**Soundness:** 1
**Presentation:** 1
**Contribution:** 2
**Rating:** 2
**Confidence:** 3

**Summary:**

The paper proposes a training framework to learn strong document representations. The framework leverages a variety of signals (text content, layout geometry, multi-view crops, layout labels) that are aggregated and combined in a multi-part loss to produce patch and page level embeddings that transfer well to downstream to shallow finetuning on DocLayNet and PubLayNet benchmarks.

**Strengths:**

Combining various signals (either obtained in a self-supervised way or through external LLM labeling) leads to strong improvements over the vanilla baseline on the evaluated tasks. The use of Multi-Positive Contrastive Learning for this task and its very positive impact on results is interesting - although (if I understand correctly) the single positive training is done by having false negatives in the CL (ie. positives that are considered negatives because only one positive can be positive) which would largely skew the results. The ablation in Table 3 is of value, showing how the different loss components and multiple signals positively combine.

**Weaknesses:**

The biggest weakness are the narrow experimental settings and results. Table 1 seems quite unclear to me (what are the baselines, what are the models in this paper). A ton of hybrid text/layout models exist (LayoutLMv2/v3/XLM) that could be evaluated here as well. The (few) baselines use reported results from the original paper and are not trained on the same data. Furthermore, benchmark results are close to 100 which could indicate benchmark saturation.

For table 2 - it is unclear what the document retrieval experimental settings are. Many document retrieval benchmarks exist nowadays, with both text-only and visual retrieval baselines (ViDoRe, ViDoRe v2, Jina VDR). The paper here provides no explanation on their evaluation settings and no baselines beyond a simple ablation. Many approaches have identified the layout/text semantics problem in recent years (from layout models LayoutLM/LILT to fully visual retrieval ColPali/DSE) and would be important to baseline in retrieval settings.

Overall, I find the paper a bit hard to read and understand. Tables are unclear, the format is not always very polished. I find the writing sometimes overly complicated, with strong claims that are often not backed by citations or experiments.  The last sentence of the conclusion is repeated almost identically twice. The abstract and conclusion claim this technique would be naturally compatible with LLMs for improved document reasoning or QA but it isn't clear to me why this would be the case, and no experiments back up this claim.

**Questions:**

- Is the SP setting training done by having false negatives in the CL (ie. positives that are considered negatives because only one positive can be positive) ? If so, why not exclude them from the loss ? If not, mentioning this would strenghten the work.

- Why do you talk about "zero" shot document retrieval ? from my understanding retrieval is evaluated in domain. Can you provide more details into how retrieval is evaluated?

- Could you clarify why and how produced embeddings would be compatible with LLMs "naturally" ? Is the claim that they would be better inputs than image embeddings used in current VLMs ? Is the necessity for an external VLM based OCR system a bottleneck here ?

> "We treat the dual-head MPCL not as an ad-hoc extension, but as a structural necessity. When
semantic and geometric cues are fused directly, they often compete and degrade retrieval; when
decoupled under multi-positive alignment, they instead provide complementary supervision."

Would be interesting to either confirm these claims experimentally, cite relevant work, or hedge the claims.

---

### Official Review · Reviewer_vYjS · 2025-10-27

**Soundness:** 2
**Presentation:** 1
**Contribution:** 2
**Rating:** 2
**Confidence:** 2

**Summary:**

This work introduces a theoretically rigorous and ingeniously designed pre-training framework for documents. By unifying multi-positive alignment and hierarchical distillation, it effectively addresses the joint understanding of document structure and semantics, demonstrating outstanding performance particularly in zero-shot retrieval, and paving the way for its integration with LLMs for advanced reasoning.

**Strengths:**

1.Two complementary alignment heads are designed: a semantic head and a geometric head. This explicitly models and aligns the content (semantics) and structural (layout) information of documents separately. This enables the model to learn representations that are both "language-grounded" and "layout-aware," allowing it to understand both "what the document says" and "how the information is organized." This is crucial for document understanding.
2.The multi-positive InfoNCE loss is adopted, addressing the limitations of "single-positive" training in traditional CLIP-style approaches. In documents, one image region may correspond to multiple text descriptions (and vice versa). This method better handles such "one-to-many" correspondences.The paper explicitly states that this alleviates "text-body bias" and significantly improves zero-shot document retrieval accuracy. This represents a substantial improvement over existing methods.

**Weaknesses:**

1、The complexity of the training objective. In addition to the benefit, the integration of dual alignment, multi-positive contrastive learning, and hierarchical self-distillation likely makes the training process considerably complex. The paper does not specify the critical factors for practical applications especially the hyper-parameter settings (lambada1-5).

2.The semantic alignment head explicitly relies on OCR-derived text spans. This means the model's performance is heavily constrained by the quality of the upstream OCR system. If the OCR fails on complex documents (e.g., handwritten text, low-quality scans, or unconventional layouts), errors could propagate and adversely affect the entire system's performance.

3. It lacks the strong or clear motivation for the adoption of the self-distillation and regularization.

4.The geometric head is described as using "box-text" descriptors to capture "class type." However, the exact meaning of "class type" remains unclear—whether it refers to physical structures (e.g., headings, paragraphs, lists) or more semantic categories. This ambiguity raises concerns about annotation costs and the model's generalization capability.

5.The experimental settings are limited. Need broader validation with more powerful general-purpose multimodal models (e.g., GPT-4V, LLaVA) and with more benchmarks such as FUNSD, DocBank

6.typos:seems the layout of this document have some issues like page 8 line 412

7. This paper needs improvement in writing, it is hard to follow with some back and forth checking the notations.

**Questions:**

See the weakness

---

### Official Review · Reviewer_6Zbp · 2025-11-01

**Soundness:** 3
**Presentation:** 2
**Contribution:** 2
**Rating:** 6
**Confidence:** 2

**Summary:**

This paper introduces GSDISTILL, a unified pretraining paradigm for document understanding. Unlike previous modular methods, it redefines document pretraining as multi-positive, layout- and semantics-aware stochastic alignment—using two complementary heads (a semantic head aligning page embeddings with OCR text spans, and a geometric head aligning with layout descriptors) trained via multi-positive InfoNCE. It also integrates a teacher-student self-distillation module with local-global regularization to improve representation quality, resulting in layout-aware, language-grounded document representations that work for tasks like retrieval and layout detection.

**Strengths:**

1. Overall, this paper is well written.

2. It redefines document pretraining as multi-positive, layout-semantics stochastic alignment (vs. fragmented heuristics) and creatively combines dual-head MPCL (semantic+geometric) with hierarchical self-distillation

3. Rigorous validation on PubLayNet/DocLayNet, ablation studies to isolate contributions ensure reproducibility.

**Weaknesses:**

1. Labels are from Gemini-2.0-flash, but the authors should explain this choice, especially there exist newer, stronger models. Also, there’s no word on data refinement (like fixing OCR errors) for these labels.

2. The 600K annotated tuples are far smaller than billion-scale vision-language datasets. This may restrict generalization, yet the paper doesn’t analyze how data size impacts performance (e.g., ablation on 100K/300K/600K tuples) or outline scaling plans (e.g., integrating public datasets); clarifying these would address concerns about real-world applicability.

3. The paper only tests layout detection, but fails to validate its claimed support for key downstream tasks (document reasoning, VQA)—a core selling point of the "unified paradigm." Without experiments on benchmarks like DocVQA or InfoVQA, the framework’s ability to enable advanced document understanding lacks evidence.

**Questions:**

See weaknesses.

---

### Note · Authors · 2025-11-12

I have read and agree with the venue's withdrawal policy on behalf of myself and my co-authors.